# Risk of atrial fibrillation and association with other diseases: protocol of the derivation and international external validation of a prediction model using nationwide population-based electronic health records

Ramesh Nadarajah [1,2] Jianhua Wu,[1,3] Ronen Arbel [4,5] Moti Haim,[6,7] Doron Zahger,[8] Talish Razi Benita,[7,9] Lior Rokach,[10] J Campbell Cowan,[2] Chris P Gale[1,2]

For numbered affiliations see end of article.

**Correspondence to**
Dr Ramesh Nadarajah;
r.nadarajah@leeds.ac.uk

## ABSTRACT

**Introduction** Atrial fibrillation (AF) is a major public health issue and there is rationale for the early diagnosis of AF before the first complication occurs. Previous AF screening research is limited by low yields of new cases and strokes prevented in the screened populations. For AF screening to be clinically and cost-effective, the efficiency of identification of newly diagnosed AF needs to be improved and the intervention offered may have to extend beyond oral anticoagulation for stroke prophylaxis. Previous prediction models for incident AF have been limited by their data sources and methodologies.

**Methods and analysis** We will investigate the application of random forest and multivariable logistic regression to predict incident AF within a 6-month prediction horizon, that is, a time-window consistent with conducting investigation for AF. The Clinical Practice Research Datalink (CPRD)-GOLD dataset will be used for derivation, and the Clalit Health Services (CHS) dataset will be used for international external geographical validation. Analyses will include metrics of prediction performance and clinical utility. We will create Kaplan-Meier plots for individuals identified as higher and lower predicted risk of AF and derive the cumulative incidence rate for non-AF cardio-renal-metabolic diseases and death over the longer term to establish how predicted AF risk is associated with a range of new non-AF disease states.

**Ethics and dissemination** Permission for CPRD-GOLD was obtained from CPRD (ref no: 19_076). The CPRD ethical approval committee approved the study. CHS Helsinki committee approval 21-0169 and data usage committee approval 901. The results will be submitted as a research paper for publication to a peer-reviewed journal and presented at peer-reviewed conferences.

**Trial registration number** A systematic review to guide the overall project was registered on PROSPERO (registration number CRD42021245093). The study was registered on ClinicalTrials.gov (NCT05837364).

## STRENGTHS AND LIMITATIONS OF THIS STUDY

⇒ Large and nationwide datasets representative of the community-dwelling populations in two countries.

⇒ Predicting the risk of incident atrial fibrillation (AF) in the short term may be more useful to screening than longer prediction horizons.

⇒ Quantification of the strength of association between predicted AF risk and other diseases may uncover other opportunities that could be actioned during AF screening beyond stroke prophylaxis.

⇒ A calculator created from a parsimonious model may enhance the usability of the model in the real world and in contexts where electronic health records are unavailable or incomplete.

⇒ It is estimated that more than a quarter of individuals living with AF are not diagnosed during routine care, which may mean that the performance of the prediction model may vary in a screening setting.

## INTRODUCTION

Atrial fibrillation (AF) is the most common sustained cardiac arrhythmia. Over the last 20 years, the number of new cases of AF diagnosed each year has risen by 72%, and now surpasses the four most common causes of cancer combined.[1] Moreover, it is estimated that up to 35% of the disease burden remains undiagnosed,[2] and 15% of strokes occur in the context of undiagnosed AF.[3]

Oral anticoagulants can reduce the risk of stroke by up to two-thirds in those with AF at higher risk of stroke,[4] and international guidelines recommend their use in patients with AF at elevated thromboembolic risk.[5] Early detection of AF may permit the initiation of oral anticoagulation to reduce embolic stroke risk,[4] and early antiarrhythmic therapy

to reduce the risk of death and stroke.[6] Accordingly, early AF detection is a key cardiovascular priority in the UK National Health Service (NHS) Long Term Plan,[7] and the European Society of Cardiology recommends opportunistic screening by pulse palpation or ECG rhythm strip in persons aged ≥65 years and systematic ECG screening in those aged ≥75 years.[8]

Furthermore, AF frequently develops due to, and in parallel with, other cardiovascular, renal and metabolic conditions,[9] and individuals with AF are at an increased risk of major cardiovascular events in excess of stroke including ischaemic heart disease, heart failure, chronic kidney disease, peripheral vascular disease and death.[10] Thus, AF screening, with or without AF diagnosis, maybe a key opportunity for holistic management of cardiometabolic risk factors and unhealthy lifestyle behaviours to reduce an individual's risk of later adverse events beyond that of stroke prophylaxis alone.

Several randomised clinical trials (RCTs) have shown that serial or continuous non-invasive ECG monitoring in older people with stroke risk factors/elevated N-terminal pro B-type natriuretic peptide, leads to a higher detection rate of previously undiagnosed AF compared with routine standard of care, though yields remain relatively low (3.0%–4.4%).[11–14] The STROKESTOP (Systematic ECG Screening for Atrial Fibrillation Among 75 Year Old Subjects in the Region of Stockholm and Halland, Sweden) RCT, where AF screening was offered to individuals aged 75 and 76 years without exclusions, achieved only a 3% yield of new AF cases with a modest benefit in a composite outcome of ischaemic or haemorrhagic stroke, systemic embolism, bleeding leading to hospitalisation and all-cause death; and not for each of ischaemic stroke, haemorrhagic stroke or hospitalisation for major bleeding.[15] Accordingly, for AF screening to be effective the yield of newly diagnosed AF among participants needs to be improved and the intervention offered may have to extend beyond only oral anticoagulation for stroke prophylaxis (figure 1).

A large proportion of the population is registered in primary care with a routinely collected electronic health record (EHR).[16 17] A prediction model that uses data available in the community to calculate AF risk could discriminate patients into risk categories, with screening offered only to higher risk individuals,[18] enabling scalable and efficient targeted AF screening. To date, several multivariable prediction models have been created or tested for the prediction of incident AF in community-based EHRs, but are of limited clinical utility for AF screening on account of moderate discriminative performance, long prediction horizons and limited scalability due to missing data.[19] None have yet reached widespread clinical practice. Moreover, reports of prediction models have yet to quantify the association between AF risk and new disease states outside that of AF and stroke.

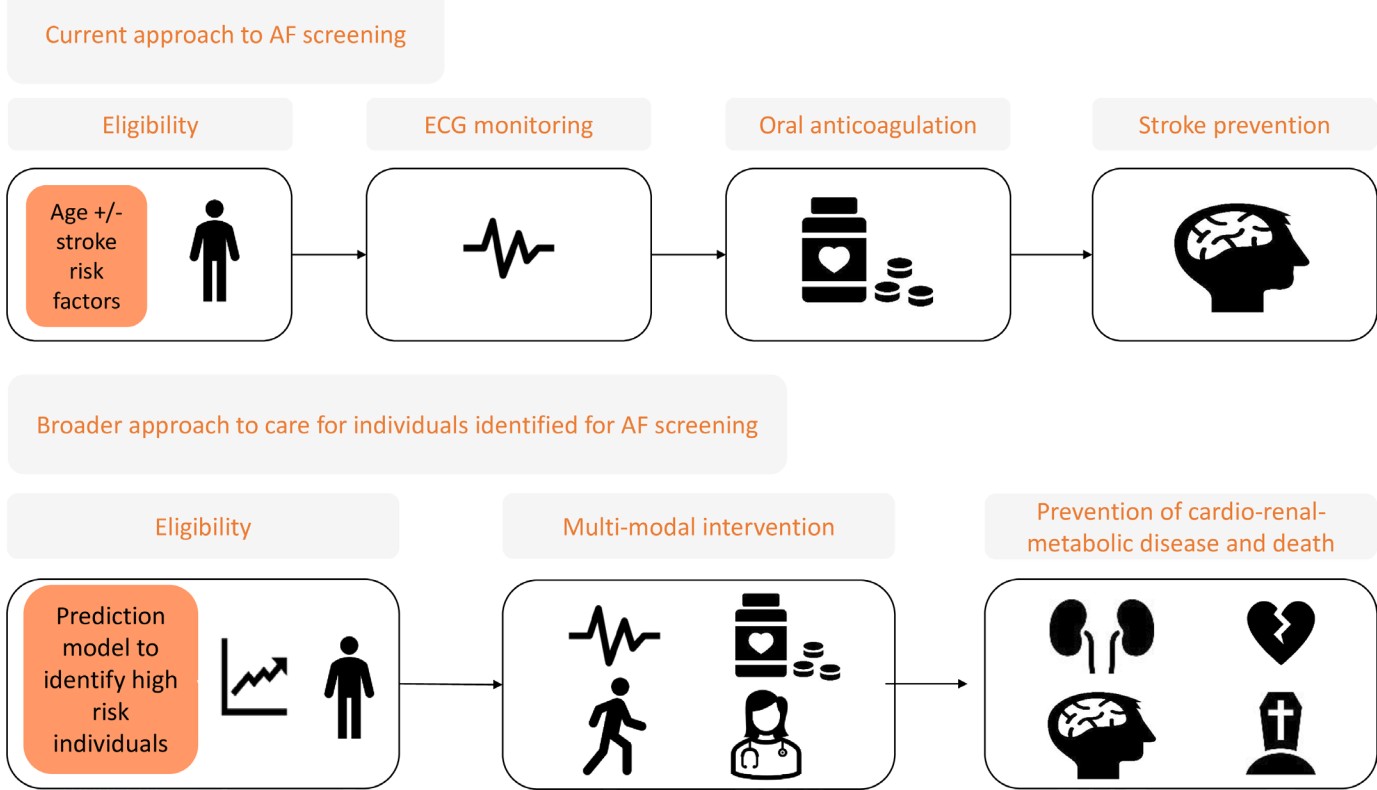

**Figure 1** A schematic representation comparing current AF screening approaches, which focus on stroke prevention, with a broader approach to AF screening that considers that individuals eligible for AF screening will be at risk of multiple outcomes beyond stroke. AF, atrial fibrillation.

## Research aim

The aims of this study are to:

1. Develop a model for predicting short-term AF risk from data routinely available in community-based EHRs.
2. Quantify the association of predicted AF risk with a range of non-AF diseases.
3. Externally validate the prediction model in an international context to assess transportability.
4. Produce a calculator derived from a parsimonious prediction model.

## METHODS AND ANALYSIS

### Data sources and permissions

The derivation dataset will be the Clinical Practice Research Datalink-GOLD (CPRD-GOLD) dataset. This is an ongoing primary care database, established in 1987, which comprises anonymised medical records and prescribing data contributed by general practices using Vision software. It contains data for approximately 17.5 million patients, with 30% of contributing practices in England, and represents the UK population in terms of age, sex and ethnicity.[16] In order to contribute to the database, general practices and other health centres must meet prespecified standards for research-quality data ('up to standard').[16 20]

Recorded information includes patients' demography, clinical symptoms, signs, investigations, diagnoses, prescriptions, referrals, behavioural factors and test results entered by clinicians, and other practice staff. All clinical information is coded using Read Codes.[21] Extracted patients will have patient-level data linked to Hospital Episode Statistics (HES), Admitted Patient Care (APC) and Office for National Statistics (ONS) Death Registration. The CPRD dataset has been used to develop or validate a range of risk prediction models.[22]

The extracted dataset, including linked data, comprises all patients for the period between 2 January 1998 and 30 November 2018 from the snapshot of CPRD-GOLD in October 2019. Over this study period, the CPRD-GOLD dataset comprises approximately 2 million patients eligible for data linkage at an up-to-standard practice, with over 200 000 patients having a record of AF during follow-up.

To ascertain whether the prediction model is transportable to geographies outside of the UK, we will externally validate its performance in the Clalit Health Services (CHS) database in Israel. As a result of the National Health Insurance Law, Israeli citizens are required to enrol in one of four payer-provider health funds and receive free basic healthcare. CHS provides health insurance coverage to 4.8 million insured members and about two-thirds of the population aged >65 years. CHS is recognised globally as the primary source of evaluation of COVID-19 vaccinations and therapies.[23–26] All clinical information is coded in International Classifications of Diseases, Ninth Revision (ICD-9). Receipt of vital status from the Ministry of the Interior ensures 100% follow-up

of mortality. We will include participants insured by Clalit with continuous membership for at least 1 year before 1 January 2019: 2 159 663 patients with 4330 of them having a new incident of AF (AF and/or atrial flutter (AFl)) in the first half of 2019.

### Patient and public involvement

The Arrhythmia Alliance and AF association provided input on the scientific advisory board for this research programme, and our patient and public involvement group have given input to reporting and dissemination plans of the research.

### Inclusion and exclusion criteria

The study population for derivation and internal validation will comprise all available patients in CPRD-GOLD eligible for data linkage and with at least 1-year follow-up in the period between 2 January 1998 and 30 November 2018. For the external validation, the study population will comprise participants insured by CHS, including those with continuous membership for at least 1 year, before 1 January 2019. Patients will be excluded if they were ≤30 years of age, or diagnosed with AF or AFl at the point of study entry, registered for less than 1 year or, in CPRD, ineligible for data linkage. Patients younger than 30 years of age are not included in the cohort for AF prediction because the incidence of AF over even a 10-year horizon is very low in this group.[1]

### Prediction model outcome ascertainment

The outcome of interest is first diagnosed AF or AFl after baseline. Baseline is taken in the CPRD-GOLD dataset as the first entry of the patient into the dataset. We have included AFl as an outcome since it has similar clinical relevance, including thromboembolic risk and anticoagulation guidelines, as AF.[5] These will be identified using Read codes in CPRD dataset. For HES APC events and underlying cause of death variables in the ONS Death Registration data file, ICD-10 codes will be used. For CHS, events will be identified using ICD-9 codes. It should be noted that a report has estimated that 305 262 individuals in the UK have undiagnosed AF,[27] and so incidence of AF within the study may be underestimated as there will be individuals with unrecorded asymptomatic AF.

### Sample size

To develop a prognostic prediction model, the required sample size may be determined by three criteria suggested by Riley et al[28] For example, suppose a maximum of 200 parameters will be included in the prediction model and the Cox-Snell generalised $R^2$ is assumed to be 0.01. A total of 377 996 patients will be required to meet Riley's criterion (1) with global shrinkage factor of 0.95; this sample size also ensures a small absolute difference ($\Delta < 0.05$) in the apparent and adjusted Nagelkerke $R^2$ (Riley's criterion (2)) and ensures precise estimate of overall risk with a margin of error<0.001 (Riley's criterion (3)). According to the Quality and Outcomes Framework, the prevalence of AF in England is 1.7%.[29 30] Given an AF prevalence of

1.7%, only 6425 patients will be expected to develop AF from 377996 patients. Within the CHS database, there are 2 159 663 patients. Therefore, the number of patients in the CPRD and CHS datasets with AF will provide sufficient statistical power to develop and validate a prediction model with predefined precision and accuracy.

### Predictor variables

A systematic review has been conducted to establish predictor variables included in varying combinations by preceding prediction models developed to detect incident AF in community-based EHRs (online supplemental table 1),[31] and supplemented with a literature search for variables associated with incident AF.

Candidate variables include
1. Sociodemographic variables including age, sex and ethnicity (SocioEconomic Score and population sector will serve as surrogate for ethnicity in CHS).
2. All disease conditions included in the patient's record, including hospitalised diseases and procedures, such as other cardiovascular diseases, diabetes mellitus, chronic lung disease, renal disease, inflammatory disease, cancer, hypothyroidism and surgical procedures.
3. Lifestyle factors including smoking status and alcohol consumption that are coded in structured Read codes.

Predictive factors will be identified using the appropriate codes, with Read codes for diagnoses and lifestyle factors. Code lists for predictors will be used from publications if available, otherwise, the CPRD code browser will be used and codes checked by at least two clinicians. The code lists for predictors in CPRD-GOLD will be adapted from CALIBER and HDR UK repositories or publications. If none are available from these sources then new code lists are developed using the OpenCodelists and checked by at least two clinicians. Diagnostic code lists will comprise the primary care coding system (Read codes), to ensure that only information readily available within a primary care EHR could be incorporated within the prediction model. Within CHS, the code lists for predictors will be developed using similar methods based on the medical records and coding of CHS, which also includes a validated chronic diseases registry.

Candidate variable data types are deliberately limited to ensure widespread applicability of the model given the reality of 'missing' data in routinely collected EHRs.[17] Observations and laboratory results are not included. Ethnicity information is routinely collected in the UK NHS and so has increasingly high completeness,[32] and we will include an 'ethnicity unrecorded' category where it is unavailable because missingness is considered informative.[33] Ethnicity in a UK context does not directly translate to an Israeli context so sociodemographic surrogates will be used: (1) population sectors—General Jewish, ultra-orthodox Jewish and Arab and (2) Socioeconomic score on a scale of 1–10. For diagnoses, if a medical code is present in the patient record (without a preceding time window limitation) then the variable is classified as being present for the patient. If medical codes are absent in a patient record we will assume that the patient does not have that diagnosis, or that the diagnosis was not considered sufficiently important to have been recorded by the GP in case of symptoms.[34] Concordantly, the analytical cohorts are not expected to have missing data for any of the predictor variables. It is possible that diagnoses may be recorded as free text, data to which we do not have access, rather than as diagnostic codes and this may lead to misclassification of some patients.

### Data analysis plan
#### Data preprocessing
The CPRD-GOLD and CHS data will be cleaned and preprocessed for model development, internal validation and external validation. Specifically, for patient features with binary values, 0 and 1 will be mapped to the binary values. Variables with multiple categories (ethnicity) will be split into their component categories, and each given a binary value to indicate the presence or not of the variable for each patient. Continuous variables (age) will be kept as continuous.

#### Descriptive analysis
Continuous variables will be reported as mean±SD and categorical variables as frequencies with corresponding percentages.

#### Prediction model development
We will compare a machine learning and logistic regression approach to prediction model development for incident AF in CPRD-GOLD. Logistic regression model offers a more manageable approach for implementation, interpretation and training compared with machine learning algorithms, but machine learning methods can better handle non-linearities and interactions among variables and may lead to better discriminative performance.[19]

We will investigate the use of a random forest (RF) classifier for AF prediction in the CPRD-GOLD dataset. In our systematic review of AF prediction in EHRs, it had the most evidence for use and showed robust performance in different datasets and geographies.[19] RF is an ensemble technique that combines a large number of decision trees using a bagging approach to improve the overall performance (figure 2).[35] In brief, the bagging approach grows multiple classification trees in parallel where each tree gives a classification which is called votes. These votes are then aggregated to provide a more accurate and stable prediction. Furthermore, the degree of variation of each feature in an RF classifier for the prediction task can be calculated using the mean decrease in the Gini coefficient, a measure of how each variable contributes to the homogeneity of nodes and leaves in the resulting RF. Showing the importance of variables used in prediction (explainability) is considered important for clinical uptake of prediction models,[36] and a limitation of using deep learning techniques. The RF model will be trained in the training dataset to predict a binary classification of developing AF or not. The model will return

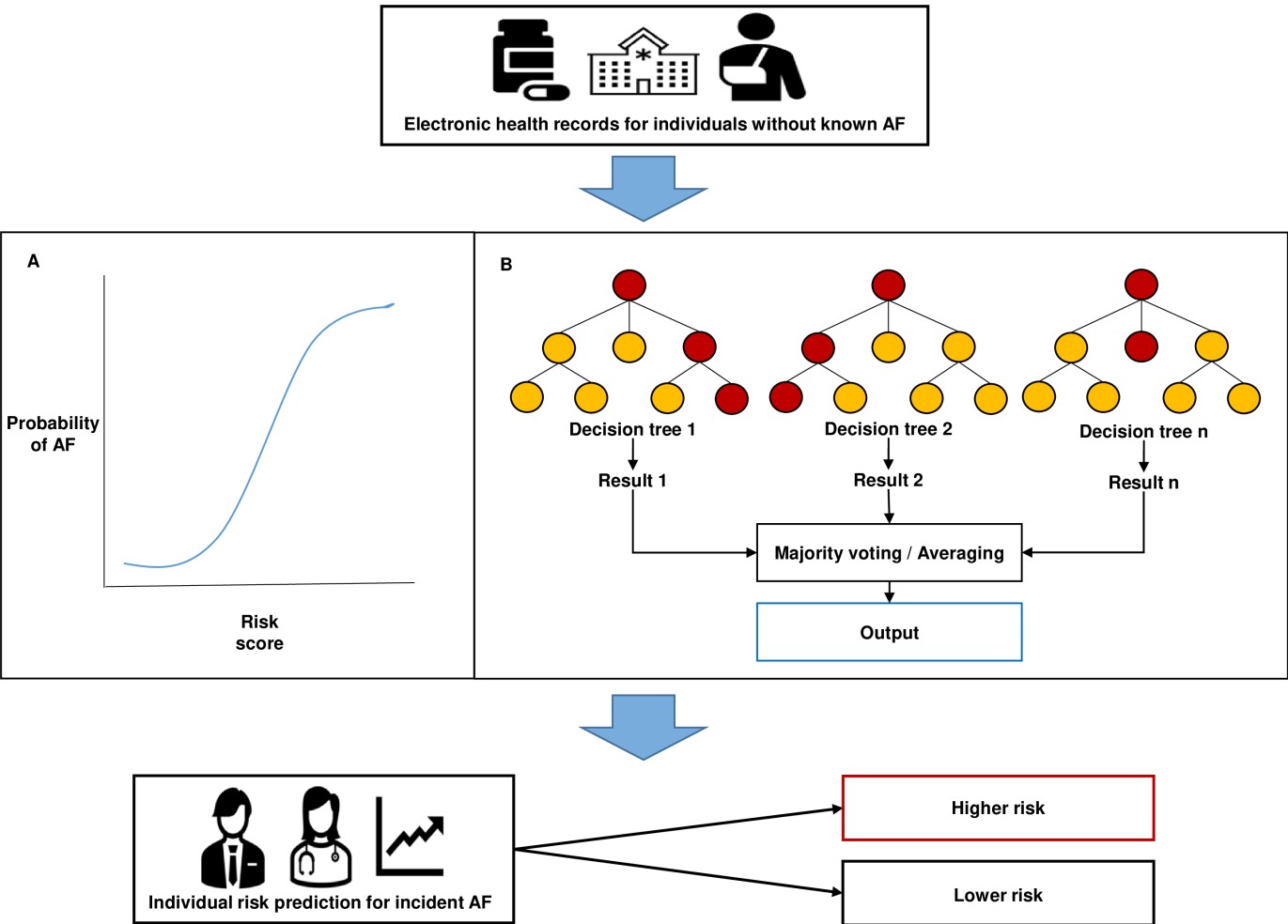

**Figure 2** A schematic representation of a multivariable logistic regression model or random forest model using data from electronic health records to provide risk prediction for incident AF. AF, atrial fibrillation.

the probability between 0 and 1 for developing AF in the training set, corresponding to the predicted probabilities of developing AF.

Preprocessed patient-level data in CPRD-GOLD will be randomly split into an 80:20 ratio to create derivation and internal validation (or training and testing) samples. The split ratio is not a significant factor, given the volume of the sample size. The model parameters and drop-out rate will be chosen through a grid search and 10-fold cross-validation will be used (ie, 10% of the training data will be randomly selected as the cross-validation set). The multivariable logistic regression model will be developed with backward model selection with Akaike information criterion.[37] The prediction window will be set at 6 months, as this is considered in keeping with the logistical time frames for organising AF investigation at scale.

### Internal validation

We will evaluate the model performance using a validation cohort with internal bootstrap validation with 200 samples. The model will be applied to the testing dataset with the same predictor variables. The area under the receiver operating characteristic curve (AUROC) will be used to evaluate predictive ability

(concordance index) with 95% CIs calculated using the DeLong method.[38] Youden's index will be established for the outcome measure as a method of empirically identifying the optimal dichotomous cut-off to assess sensitivity, specificity, positive predictive value and negative predictive value. We will calculate the Brier score, a measure of both discrimination and calibration, by taking the mean squared difference between predicted probabilities and the observed outcome. To assess the clinical impact of using prediction model as opposed to other risk prediction scores, we will calculate the case reclassification, non-case reclassification and overall net reclassification index at the risk threshold that equates to the average 6 months incidence rate in the cohort and conduct a decision curve analysis, which assesses across threshold probabilities whether the predictive model would do more benefit than harm. Calibration will be assessed graphically by plotting predicted AF risk against observed AF incidence and quantified using a calibration slope.

The same methods will be employed in subgroups by age (<65 years, ≥65 years, <75 years, ≥75 years), sex (women, men) and ethnicity (white, black, Asian, others

and unspecified) to assess the model's predictive performance across clinically relevant groups.

The performance of the prediction model will be compared with the $CHA_2DS_2$-VASc and $C_2HEST$ scores. The $CHA_2DS_2$-VASc score was originally developed to predict stroke risk in individuals with AF, and the $C_2HEST$ score for Asian people without structural heart disease.[19] These algorithms are robust to missing data in routinely collected primary care EHRs and have been tested for AF risk prediction in European cohorts.[19] Other algorithms that can only be applied to a minority of European primary care EHRs (Pfizer-AI, CHARGE-AF) will not be considered as they cannot be implemented at scale to inform AF screening.[17 27]

### Quantification of the association between short-term predicted AF risk and long-term AF and other diseases

We will include all patients randomly assigned to the testing dataset in CPRD-GOLD by the Mersenne twister pseudorandom number generator, categorised as lower or higher predicted AF risk by the developed prediction model at baseline (point of entry to the study). For long-term AF risk, we will plot Kaplan-Meier plots for individuals identified as higher and lower predicted risk of AF to assess the event rate for AF censored at 10 years, and calculate the hazard ratio (HR) for AF between higher and lower predicted risk of AF using the Cox proportional hazard model with adjustment for the competing risk of death. This will inform us of whether short-term AF risk is also associated with long-term AF risk, and whether an individual who undergoes risk-guided AF screening should be considered for repeated AF screening at a later time point (eg, 1 or 5 years).

For non-AF disease states, we will consider the initial presentation of a cardiovascular, renal or metabolic disease or death. This is because AF is not a disease in isolation and is known to be associated with a high risk of adverse clinical outcomes. To best characterise highly prevalent and morbid diseases, associated with the development or consequence of AF and that may be appropriate for prevention or targeted diagnostic pathways subsequent to AF screening (online supplemental figure 1),[9] we will individually examine the following nine conditions: heart failure, valvular heart disease (and specifically aortic stenosis), myocardial infarction, stroke (ischaemic and haemorrhagic) or transient ischaemic attack, peripheral vascular disease, chronic kidney disease, diabetes mellitus and chronic obstructive pulmonary disease. These disease states have been further selected for investigation because interventions could be implemented and/or tested to reduce their clinical progression. We will also quantify the occurrence of death by any cause recorded in primary care or by death certification from the UK Death Register of the ONS, which will be mapped on to nine disease categories (online supplemental table 2). For each condition, a list of diagnostic codes from the CALIBER code repository, including from ICD-10th revision (used in secondary care) and Read coding schemes

(used in primary care) will be defined comprehensively to identify diagnoses from EHRs. Incident diagnoses will be defined as the first record of that condition in primary or secondary care records from any diagnostic position. For the definition of new cases, we will exclude individuals for the analysis of each condition who had a diagnosis of that condition before the patient's entry to the study. If no indication of a specific disease is recorded, then the patient will be assumed to be free from the disease. CPRD is a positive recording dataset, which reduces the likelihood of the non-recording of a clinically identified disease state.

We will create Kaplan-Meier plots for individuals identified as higher and lower predicted risk of AF to assess the event rate for non-AF outcomes censored at 10 years. We will derive the cumulative incidence rate for each outcome at 1, 5 and 10 years considering the competing risk of death, as well as death at 5 and 10 years. For each specified outcome, we will calculate the HR between higher and lower predicted risk of AF using Fine and Gray's model with adjustment for the competing risk of death. We will also report adjusted HR where the model is adjusted for age, sex, ethnicity and the presence of any of the other outcomes at baseline. As some of the outcomes have incidence rates that are strongly associated with age (eg, aortic stenosis) or differ by sex (eg, heart failure),[39 40] we will conduct subgroup analyses of incidence rates for higher and lower risk individuals for each outcome by age group (30–64 years and ≥65 years) and sex. As some of the non-AF outcomes are more likely to occur in the setting of prevalent AF (eg, stroke or heart failure),[9] we will also conduct a sensitivity analysis whereby people with incident AF during follow-up are excluded.

### External validation

The CHS dataset will then be used to externally validate the model performance to assess transportability. A lack of external validation hampers the implementation of prediction models in routine clinical practice.[41] The prediction model will be saved, including the RF structure, predictor variables and outcome variable into a standalone file. The file will be passed onto the external international collaborators so that they can apply the model to their local external cohort to generate predicted probability of experiencing AF at 6 months for each patient. Then the predicted probability will be compared against the observed outcome in the external cohort to assess the performance of the model. Prediction performance will be quantified by calculating the AUROC, Brier score and by using calibration plots, and the same aforementioned clinical utility and subgroup analysis will be conducted. The performance of the prediction model will be compared with the $CHA_2DS_2$-VASc and $C_2HEST$ scores.

### Prediction model calculator

The full models are developed to take advantage of rich longitudinal community-based EHRs present in many

high-income countries. However, there are other geographies (low-lower-middle-income countries) and care settings (emergency care, secondary care clinics) where searching for AF may be desired and an easy-to-use, simple model is preferable. From the derived prediction model, we will generate a parsimonious model based on factors with clinical rationale to predict new-onset AF over a 6 months time horizon.[9] This will be based on the same core principles as detailed above, but use logistic regression to ensure transparency in how prediction results are calculated. We will aim to develop a user-friendly version of a model that may be applied as a calculator in a clinical and public setting, yet have good model performance indices.

## Software

All analyses will be conducted through R.

## ETHICS AND DISSEMINATION

The study has been approved by CPRD (ref no: 19_076). Those handling data have completed University of Leeds information security training. All analyses will be conducted in concordance with the CPRD study dataset agreement between the Secretary of State for Health and Social Care and the University of Leeds.

The CHS Community Helsinki Committee and the CHS Data Utilisation Committee approved the study. The study was exempt from the requirement to obtain informed consent.

The study has been registered at ClinicalTrials.gov (NCT05837364). The study is informed by the Prognosis Research Strategy (PROGRESS) and CODE-EHR best-practice frameworks and recommendations.[41 42] The subsequent research papers will be submitted for publication in a peer-reviewed journal and will be written following Transparent reporting of a multivariable prediction model for individual prognosis or diagnosis (TRIPOD) and REporting of studies Conducted using Observational Routinely-collected Data (RECORD).[43 44]

If the model shows better prediction performance than previous models and evidence for clinical utility in analysis, it could be made readily available through EHR platforms. If the parsimonious model shows good prediction performance, the user-friendly version could be accessible through the internet. Future research would be needed to assess the clinical impact of this risk model. At the point when utilisation in clinical practice is possible, the applicable regulation on medicine devices will be adhered to.[45] When in clinical use, the model itself could also be reviewed and updated after incorporating evidence from the curation of more data.

## Author affiliations

[1]Leeds Institute of Data Analytics, University of Leeds, Leeds, UK
[2]Department of Cardiology, Leeds Teaching Hospitals NHS Trust, Leeds, UK
[3]Wolfson Institute of Population Health, Queen Mary University of London, London, UK
[4]Health Systems Management, Ben-Gurion University of the Negev, Beer-Sheva, Israel
[5]Sapir College, Sderot, Israel
[6]Department of Cardiology, Soroka University Medical Center, Beer Sheva, Israel
[7]Ben-Gurion University of the Negev, Beer-Sheva, Israel
[8]Soroka University Medical Center, Beer Sheva, Israel
[9]Clalit Health Services, Tel Aviv, Israel
[10]Department of Information Systems and Software Engineering, Ben-Gurion University of the Negev, Beer-Sheva, Israel

**Contributors** RN, JW and CG conceived the concept and planned the analysis. RN wrote the first draft, with contributions from all authors. All authors (RN, JW, CC, RA, DZ, MH, TRB, LR and CG) approved the final version and jointly took responsibility for the decision to submit the manuscript to be considered for publication.

**Funding** RN is supported by the British Heart Foundation Clinical Research Training Fellowship (FS/20/12/34789). JW is supported by Barts Charity (MGU0504). The analysis in Clalit Health Services is funded by Israel Science Foundation Precision Membership Partnership (Grant #3543/21).

**Competing interests** None declared.

**Patient and public involvement** Patients and/or the public were involved in the design, or conduct, or reporting, or dissemination plans of this research. Refer to the Methods and analysis section for further details.

**Patient consent for publication** Not applicable.

**Provenance and peer review** Not commissioned; externally peer reviewed.

**ORCID iDs**
Ramesh Nadarajah http://orcid.org/0000-0001-9895-9356
Ronen Arbel http://orcid.org/0000-0002-6058-8665

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
