## [Reviewer comments · BMJ Open]

ARTICLE DETAILS

TITLE (PROVISIONAL)	Risk of atrial fibrillation and association with other diseases: protocol of the derivation and international external validation of a prediction model using nationwide population-based electronic health records
AUTHORS	Nadarajah, Ramesh; Wu, Jianhua; Arbel, Ronen; Haim, Moti; Zahger, Doron; Benita, Talish Razi; Rokach, Lior; Cowan, Campbell; Gale, Chris

VERSION 1 – REVIEW

REVIEWER	Poppe, K University of Auckland, biostats
REVIEW RETURNED	18-Jun-2023

GENERAL COMMENTS	The protocol describes a study to predict the risk of incident AF within a 6 month prediction horizon. Then, using that information, investigate the cumulative incidence for non-AF cardio-renal-metabolic diseases and death over a longer term. Please clarify: Baseline risk factors/predictor variables: • What period of time will each person's 'baseline' RFs be defined over, ie. not all factors will have been assessed at one patient visit so how much leeway is going to be allowed? • When describing the predictor variables it is written that candidate variables include "all disease conditions during follow-up.." If a condition appears after baseline, it can't be used as a predictor. Please clarify. CPRD-GOLD data from 2 Jan1998 to 30 Nov 2018 is being accessed. Within that, what date range will define the baseline, and what date will AF need to have occurred by to start the longer term follow-up for non-AF conditions. What will be the minimum and maximum duration of follow-up available for non-AF outcomes? The study relies on extracting Read coded conditions. What % of conditions and/or patients are expected to have conditions that have only be coded as text? Where Read codes are missing or incorrect. Would this lead to any bias? Internal validation will be assessed using "... Brier score, mean squared difference between predicted probabilities and the observed outcome". How are you going to get predicted probabilities from the random forest model? Related, is how will the random forest model be applied to an
---

	external cohort? ie. what is the output of the RF model that enables it to be applied to an external cohort? Suggest broadening “calculate the NRI” to doing a NR analysis, ie. examine the components of the NRI, being reclassification of those who developed AF and of those who didn’t. Why will the models from this study be compared to CHADs2VASc, which was developed in a different type of patient cohort and predicts a different outcome?? Wouldn’t QRISK or equivalent be more or just as likely to predict onset of CVD than AF? It would also seem more compatible with your aims of using the predicting-AF algorithm to feed into the models for non-AF CVD. The acronym of FIND-AF is introduced in the Patient and Public involvement statement but it doesn’t appear to have been mentioned before that.
--	--

REVIEWER	Marwick, Tom Baker IDI Heart and Diabetes Institute
REVIEW RETURNED	01-Aug-2023

GENERAL COMMENTS	This is a well-presented study design paper. The analysis strategy is novel and interesting. My concerns are;  1) The model is based on prediction of diagnosed AF. But paroxysmal AF can be hard to diagnose and just as important to detect. The whole process of verifying AF receives inadequate attention in the paper, including the limitations. 2) I really didn't understand the reference to non-NOAC aspects of management, especially related to AF prevention (lifestyle etc). They are unquestionably important but I think peripheral to what you are proposing about AF detection. 3) Inherent in this project is the notion that large numbers are inherently beneficial in comparison with the scores created till now. I don't find this necessarily the case. In fact there has been a recent UK biobank derived score that isn't much better than CHARGE-AF (DOI:10.1093/eurheartj/ehad375). 4) In the sample calculations, a prevalence of 1.7% is mentioned. I'm unclear how your approach will distinguish between known AF at baseline vs AF that's detectable by screening 5) Failure to include observations and lab results could be a material problem. Exercise, weight and activity are important aspects. 6) I have difficulty in understanding why CHARGE-AF (age, race, smoking, MI, HF, DM, anti-HT meds) cannot be implemented at scale?
--

VERSION 1 – AUTHOR RESPONSE

Reviewer: 1
Dr. K Poppe, University of Auckland

Comments to the Author:

The protocol describes a study to predict the risk of incident AF within a 6 month prediction horizon. Then, using that information, investigate the cumulative incidence for non-AF cardio-renal-metabolic diseases and death over a longer term.

Author reply

We thank the Reviewer for their careful review of the manuscript and helpful suggestions.

Please clarify:

Baseline risk factors/predictor variables:

- What period of time will each person's 'baseline' RFs be defined over, ie. not all factors will have been assessed at one patient visit so how much leeway is going to be allowed?

Author reply

The period of time that a person's baseline risk factors are defined over is a diagnosis at any time previously recorded in their electronic health record. Primary care records are longitudinal in the UK, and thus previously recorded diagnoses can extend over the lifetime a patient has lived up to that point. That is to say, a heart failure diagnosis is counted whether it is made 6 months before the risk stratification, 6 years before or longer.

Manuscript change

For diagnoses, if a medical code is present in the patient record (without a preceding time window limitation) then the variable is classified as being present for the patient.

When describing the predictor variables it is written that candidate variables include "all disease conditions during follow-up.." If a condition appears after baseline, it can't be used as a predictor. Please clarify.

Author reply

We thank the Reviewer for spotting this error. We have corrected this.

Manuscript change

All disease conditions included in the patient's record

CPRD-GOLD data from 2 Jan1998 to 30 Nov 2018 is being accessed. Within that, what date range will define the baseline, and what date will AF need to have occurred by to start the longer term follow-up for non-AF conditions.

Author reply

Baseline is taken as when an eligible individual (aged ≥ 30 years, without previously diagnosed atrial fibrillation (AF) or atrial flutter in their record, registered for ≥ 1 year or and eligible for data linkage) enters the cohort. An individual may enter the dataset for the first time at any point during the study period.

For the quantification of the association between short-term predicted AF risk and long-term AF and other diseases, in the holdout testing dataset individuals at point of entry into the cohort will have risk score calculated based on their age, sex, ethnicity, and the presence of diagnoses in their record. Follow-up will occur from the point of risk stratification for individuals in the higher and lower predicted AF risk cohorts, irrespective of whether they are diagnosed with AF. Our hypothesis is that individuals at risk of AF are also at risk of other conditions beyond AF, and whether AF is diagnosed or not. If we demonstrate that is the case, it may suggest that individuals identified for risk-guided AF screening may benefit from being offered interventions beyond a period of ECG monitoring.

Manuscript change

Prediction model outcome ascertainment

The outcome of interest is first diagnosed AF or AFI after baseline. Baseline is taken in the CPRD-GOLD dataset as the first entry of the patient into the dataset.

Quantification of the association between short-term predicted AF risk and long-term AF and other diseases

We will include all patients randomly assigned to the testing dataset in CPRD-GOLD by the Mersenne twister pseudorandom number generator, categorized as lower or higher predicted AF risk by the developed prediction model at baseline (point of entry to the study).

What will be the minimum and maximum duration of follow-up available for non-AF outcomes?

Author reply

For non-AF outcomes we will assess the event rate censored at 10 years, thus the same time window as is used for the long-term AF outcome.

Manuscript change

We will create Kaplan-Meier plots for individuals identified as higher and lower predicted risk of AF to assess the event rate for non-AF outcomes censored at 10 years.

The study relies on extracting Read coded conditions. What % of conditions and/or patients are expected to have conditions that have only be coded as text? Where Read codes are missing or incorrect. Would this lead to any bias?

Author reply

CPRD is a positive reporting database. Therefore, if a disease code is not present it is assumed to not have been diagnosed. Validation studies have examined the validity of diagnoses coded within the database, as opposed to trying to elicit how many individuals have a diagnosis that was not coded using structured codes. Thus, we do not have data for what % of conditions or patients may have diagnoses recorded in free text but not with a structure read code. We have added this limitation.

Manuscript change

It is possible that diagnoses may be recorded as free text, data to which we do not have access, rather than as diagnostic codes and that this may lead to misclassification of some patients.

Internal validation will be assessed using “.. Brier score, mean squared difference between predicted probabilities and the observed outcome”. How are you going to get predicted probabilities from the random forest model?

Author reply

We will train the random forest model on the training dataset to predict a binary classification of developing AF or not. The model will return the probability between 0 and 1 for developing AF in the training set, then the model will be applied to the testing dataset with the same predictor variables. The output of the model is the predicted probabilities of developing AF.

Manuscript change

Prediction model development

The RF model will be trained in the training dataset to predict a binary classification of developing AF or not. The model will return the probability between 0 and 1 for developing AF in the training set, corresponding to the predicted probabilities of developing AF.

Internal validation

The model will be applied to the testing dataset with the same predictor variables.

Related, is how will the random forest model be applied to an external cohort? ie. what is the output of the RF model that enables it to be applied to an external cohort?

Author reply

Once we develop the random forest model, we will save the final model including the random forest structure, predictor variables, and outcome variable into a standalone file. The file will be passed onto external international collaborators, so that they can apply the random forest model to their local external cohort to generate predicted probability for each patient. Then the predicted probability will be compared against the observed outcome in the external cohort to assess the performance of the model.

Manuscript change

The prediction model will be saved, including the random forest structure, predictor variables, and outcome variable into a standalone file. The file will be passed onto the external international collaborators, so that they can apply the model to their local external cohort to generate predicted probability of experiencing AF at 6 months for each patient. Then the predicted probability will be compared against the observed outcome in the external cohort to assess the performance of the model.

Suggest broadening “calculate the NRI” to doing a NR analysis, ie. examine the components of the NRI, being reclassification of those who developed AF and of those who didn’t.

Author reply

We agree with the Reviewer and have broadened the description of the net reclassification analysis.

Manuscript change

To assess the clinical impact of utilising FIND-AF as opposed to other risk prediction scores, we will calculate the case reclassification, non-case reclassification, and overall net reclassification index at the risk threshold that equates to the average 6 months incidence rate in the cohort and conduct a decision curve analysis, which assesses across threshold probabilities whether the predictive model would do more benefit than harm.

Why will the models from this study be compared to CHADS₂VASc, which was developed in a different type of patient cohort and predicts a different outcome?? Wouldn't QRISK or equivalent be more or just as likely to predict onset of CVD than AF? It would also seem more compatible with your aims of using the predicting-AF algorithm to feed into the models for non-AF CVD.

Author reply

Risk prediction models for incident AF prediction applicable in the community were summarised in a systematic review and meta-analysis in 2020 (PMID: 32011689) and, specifically for electronic health records, in 2022 (PMID: 34607811).

In both of these systematic reviews the CHA₂DS₂-VASc score was found to have statistically significant summary discrimination performance for incident AF across multiple cohorts, and very

similar performance to the CHARGE-AF and FHS-AF risk scores, which had been specifically designed for incident AF prediction. QRISK has not been tested for the purpose of predicting risk of incident AF.

The aim of this study is to optimise prediction of incident AF using routinely collected data to make AF screening more efficient. The second hypothesis is that individuals at higher risk of AF will also be at higher risk of other conditions. Therefore the primary aim is to predict AF, and as such the CHA₂DS₂-VASc and C₂HES_T scores, which have been tested for prediction of incident AF risk in European cohorts, are used for comparison.

The acronym of FIND-AF is introduced in the Patient and Public involvement statement but it doesn't appear to have been mentioned before that.

Author reply

We thank the Reviewer for noting this oversight, and we have removed the acronym from the manuscript.

Reviewer: 2

Dr. Tom Marwick, Baker IDI Heart and Diabetes Institute

Comments to the Author:

This is a well-presented study design paper. The analysis strategy is novel and interesting.

Author reply

We thank the Reviewer for their careful review of the manuscript and helpful suggestions.

My concerns are;

The model is based on prediction of diagnosed AF. But paroxysmal AF can be hard to diagnose and just as important to detect. The whole process of verifying AF receives inadequate attention in the paper, including the limitations.

Author reply

We agree with the Reviewer that the shortfalls in diagnosis of AF in routine practice should be further discussed, and we have now added sections specifically about this.

Manuscript change

Strengths and limitations of the study

It is estimated that more than a quarter of individuals living with AF are not diagnosed during routine care, which may mean that the performance of the prediction model may vary in a screening setting.

Prediction model outcome ascertainment

It should be noted that a report has estimated that 305 262 individuals in the UK have undiagnosed AF,²⁸ and so incidence of AF within the study may be underestimated as there will be individuals with unrecorded asymptomatic AF.

I really didn't understand the reference to non-NOAC aspects of management, especially related to AF prevention (lifestyle etc). They are unquestionably important but I think peripheral to what you are proposing about AF detection.

Author reply

We agree with the Reviewer that hitherto AF screening has been focussed on stroke prevention to avert stroke.

A meta-analysis of four published randomised clinical trials with a total of 35 836 participants following the intention-to-treat principle demonstrated a modest point estimate in favour of AF screening reducing stroke risk (RR 0.91, 95% CI 0.84-0.99) but published trials were heterogeneous in their populations, definition of stroke, and screening methodology (PMID: 35919582). The trial sequential analysis in the meta-analysis showed that the cumulative z-score from published data is insufficient to conclude the benefits of screening and calculated an optimal sample size of a total of 103 454 participants randomised.

Thus, it is possible a large numbers of patients need to be studied to demonstrate the efficacy of AF screening for stroke prevention. Moreover, AF is only one of the risk factors of stroke, and the rate of ischaemic stroke is decreasing (PMID: 35923806).

Accordingly, it may be prudent to consider what may be the benefits of AF screening beyond prevention of ischaemic stroke. Further possible benefits from AF screening would be lower mortality and a possibility to address undetected structural heart disease and untreated cardiovascular risk factors such as hypertension, obesity, alcohol consumption and obstructive sleep apnoea (PMID: 33411987).

The minority of patients with AF die as a result of stroke (PMID: 26330425), and so it is argued that interventions aimed at reducing outcomes beyond stroke are warranted in patients with AF. Management of comorbidities and risk factors is now a central pillar of management for patients with AF. That is, a reduction of the burden of non-stroke events in individuals with AF may be actionable through a focus on the management of cardiovascular risk factors, optimisation of established cardiovascular disease, and the identification of undetected cardio-renal-metabolic disease. Our hypothesis is that individuals deemed eligible for risk-based AF screening also have risk factors, and both detected and undetected comorbidities that could be optimised to reduce the risk of events beyond stroke, irrespective of whether AF is detected during screening. That is, we are seeking to understand whether the optimum intervention for individuals identified for AF screening may extend beyond the provision of ECG monitoring for arrhythmia detection alone.

Inherent in this project is the notion that large numbers are inherently beneficial in comparison with the scores created till now. I don't find this necessarily the case. In fact there has been a recent UK biobank derived score that isn't much better than CHARGE-AF (DOI:10.1093/eurheartj/ehad375).

Author reply

We agree with the Reviewer that prediction models developed in large datasets will not inherently be better than models derived from smaller datasets, and the Reviewer is correct in their summation of the HARMS2-AF paper they referenced. Future work could look to evaluate the model developed against models developed in smaller datasets in other settings.

In the sample calculations, a prevalence of 1.7% is mentioned. I'm unclear how your approach will distinguish between known AF at baseline vs AF that's detectable by screening

Author reply

We start with a cohort without AF (a past medical history of AF or atrial flutter is an exclusion criteria).

We used the prevalence of 1.7% in the sample size calculation to estimate the number of AF incidence we need to observe in order to achieve the desired power. We demonstrated that given the total size of the UK and external datasets, we would have enough number of patients and cases to develop a prediction model and then externally validate it.

Failure to include observations and lab results could be a material problem. Exercise, weight and activity are important aspects.

I have difficulty in understanding why CHARGE-AF (age, race, smoking, MI, HF, DM, anti-HT meds) cannot be implemented at scale?

Author reply

We agree with the Reviewer that observations such as height, weight and BMI, and laboratory results such as BNP, can be strongly associated with risk of incident AF, and that these have been utilised in previous prediction models for incident AF (PMID: 32011689, PMID: 34607811). CHARGE-AF incorporates age, race, height, weight, systolic, diastolic blood pressure, smoking, use of an anti-hypertensive medication, diabetes, heart failure, and myocardial infarction (PMID: 23537808).

However in routinely collected primary care data observations are often missing. Within our CPRD-GOLD dataset systolic and diastolic blood pressure, heart rate and body mass index are missing for 17.0%, 17.0%, 70.3%, and 53.4%, respectively of participants within one year of cohort entry.

Furthermore, when a research group tried to implement CHARGE-AF in routinely-collected primary care data in the Netherlands for individuals aged ≥ 40 years without a history of AF, they found they could only apply the model to 17.2% of the cohort because of missing data.

We aim to develop a model that can be implemented at scale within existing primary care electronic health record systems remotely without the requirement for individuals to attend a further appointment to collect data. Therefore we are not including observations and laboratory results in the model development protocol.

VERSION 2 – REVIEW

REVIEWER	Poppe, K University of Auckland, biostats
REVIEW RETURNED	17-Sep-2023
GENERAL COMMENTS	Thank you for addressing my comments. It is an interesting study, particularly as the team have thought beyond risk of stroke to what a broader prediction can mean for both the patient and healthcare delivery.